# Is There a Role for Gut Microbiome Dysbiosis in IgA Nephropathy?

**DOI:** 10.3390/microorganisms10040683

**Published:** 2022-03-22

**Authors:** Renato C. Monteiro, Dina Rafeh, Patrick J. Gleeson

**Affiliations:** 1NIH Department, Faculté de Médecine, Université de Paris Cité, 75018 Paris, France; diina.rafeh@gmail.com (D.R.); james.gleeson@inserm.fr (P.J.G.); 2INSERM U1149, Centre de Recherche sur l’Inflammation, 75018 Paris, France; 3CNRS ERL8252, 75018 Paris, France

**Keywords:** IgA nephropathy, microbiome, dysbiosis, IgA1 glycosylation, CD89, FMT

## Abstract

Immunoglobulin A nephropathy (IgAN) is the most common primary glomerulonephritis and one of the leading causes of renal failure worldwide. The pathophysiology of IgAN involves nephrotoxic IgA1-immune complexes. These complexes are formed by galactose-deficient (Gd) IgA1 with autoantibodies against the hinge region of Gd-IgA1 as well as soluble CD89, an immune complex amplifier with an affinity for mesangial cells. These multiple molecular interactions result in the induction of the mesangial IgA receptor, CD71, injuring the kidney and causing disease. This review features recent immunological and microbiome studies that bring new microbiota-dependent mechanisms developing the disease based on data from IgAN patients and a humanized mouse model of IgAN. Dysbiosis of the microbiota in IgAN patients is also discussed in detail. Highlights of this review underscore that nephrotoxic IgA1 in the humanized mice originates from mucosal surfaces. Fecal microbiota transplantation (FMT) experiments in mice using stools from patients reveal a possible microbiota dysbiosis in IgAN with the capacity to induce progression of the disease whereas FMT from healthy hosts has beneficial effects in mice. The continual growth of knowledge in IgAN patients and models can lead to the development of new therapeutic strategies targeting the microbiota to treat this disease.

## 1. The IgA System and Mucosal Immunity

Immunoglobulin A (IgA) is the most important antibody class in mucosal secretions. Adult humans transport some 3 g/d of IgA into external secretions, more than the daily production of all other Ig isotypes combined [1,2]. Moreover, IgA has a significant serum concentration (about 2–3 mg/mL), being the second most prevalent antibody in the serum after IgG [2]. IgA is differentially distributed between the systemic and mucosal immune systems and plays a key role in immune protection. Circulating IgA are mainly monomers derived from bone marrow plasma cells, whereas the majority of IgA in the secretions are polymeric, mainly in the form of dimers comprising two IgA monomers linked by a J (joining) chain [3]. Dimeric IgA is synthesized by local plasma cells before being transported to mucosal surfaces through epithelial cells by the polymeric Ig receptor (pIgR) [4]. After protease cleavage of polymeric Ig receptor extracellular domains, also named as secretory component (SC), IgA is released into secretions associated with the SC so-called Secretory IgA (SIgA). Human IgA is divided into two closely related subclasses, IgA1 and IgA2, which differ by the absence of a thirteen-amino acid sequence in the hinge region of the IgA2 molecule [2,3]. This missing sequence contains a specific zone for IgA1 protease cleavage which explains the resistance of IgA2 to the action of bacterial proteases and may underlie the predominance of IgA2 in mucosal secretions [2]. Interestingly, while two IgA subclasses are recognized in humans, only one class exists in rodents. Serum monomeric IgA (mIgA) is thought to play a minor role in systemic immune responses. The major role of serum mIgA in physiology is to promote a powerful anti-inflammatory effect. It has been demonstrated by several groups more than 30 years ago that in the absence of antigen, serum IgA is capable of down-regulating many cell responses [5]. The molecular basis for IgA anti-inflammatory function action was found to be at least in part mediated by the CD89, an IgA Fc receptor [6]. CD89 (or FcαRI) is a transmembrane protein composed of two extracellular domains, a transmembrane domain, and an intracytoplasmic tail. Its expression is restricted to cells of the myeloid lineage [3]. CD89 requires an associated adaptor, the FcRγ chain to transduce signals. CD89 is a unique member of the FcR family its gene is not located in the FcR gene cluster but on chromosome 19, inside the leukocyte receptor cluster (LRC). CD89 is distantly related to other FcRs, being more homologous to LRC-encoded activating and inhibitory receptors. Inhibitory CD89 signals delivered by serum monomeric IgA (mIgA) are important to control the immune system by preventing the development of autoimmunity and inflammation through receptor-mediated phosphatase recruitment [7]. Moreover, supporting evidence for the inhibitory role of serum monomeric IgA also comes from patients with selective IgA deficiency as they can develop inflammatory and auto-immune diseases [8]. Polymeric IgA antibodies participate in immune responses inducing activation of eosinophils, neutrophils, monocytes and macrophages leading to several protective functions, such as cytokine release, phagocytosis and ADCC [3]. CD89 also allows IgA internalization in myeloid circulating cells, where they are catabolized. Thus dual CD89 function redefines this receptor as a molecular switch of the immune system, directing signals towards either an inhibitory or an activating function [9].

The respiratory, digestive and urogenital tracts of the human body are lined by selectively permeable barriers, the mucus membranes, which separate the internal body organs from the external milieu. The gut is equipped with a mucosal immune system that defends against invading pathogens and maintains immunological homeostasis along the epithelium. The effectors involved in this include the mucus layer (comprising mucin secreted by the intestinal goblet cells, antimicrobial peptides and SIgA), the underlying epithelial cells with tight junctions in between them, and the lamina propria containing innate and adaptive immune cells (macrophages, dendritic cells, B cells, and T cells…) [10]. SIgA secreted into the lumen patrol the mucus and entrap dietary antigens and microorganisms. SIgA has a particular feature of promoting the uptake and delivery of luminal antigens to the DCs located in gut associated lymphoid tissues (GALT), influencing inflammatory responses via a process known as reverse transcytosis [11]. This process is mediated through specialized epithelial cells scattered along the intestinal epithelium known as M cells. M cells internalize the Ag-SIgA complex via the Dectin-1 receptor which has been found to be necessary for the reverse transcytosis pathway [12]. The dendritic cells process and present the antigen to CD4+ T cells which will promote the IgA-producing B cells. The high-affinity antibodies produced from this T cell-dependent pathway are thought to protect intestinal mucosal surfaces against invasion by pathogenic microorganisms by exclusion mechanism. In addition, the dendritic cells and the epithelial cells release the B cell activation factor of the TNF family (BAFF), APRIL, and IL-10 cytokines in the environment which favor the induction of IgA-producing B cells [13]. These low-affinity IgA antibodies (from T cell-independent pathways) are important to confine commensal bacteria to the intestinal lumen [14,15].

While high-affinity IgA antibodies (from T cell-dependent pathways) are thought to protect intestinal mucosal surfaces against invasion by pathogenic microorganisms by exclusion mechanism, low-affinity IgA antibodies (from T cell-independent pathways) are important to confine commensal bacteria to the intestinal lumen [14,15].

As the largest micro-ecological system of the human body, the intestinal tract mainly includes intestinal bacteria, in addition to a very small amount of viruses, mycoplasmas and fungi. The intestinal tract harbors more than 1000 bacterial species, constituting over 10 times the total number of human cells [16]. Therefore, homeostasis of the intestinal flora is physiologically significant in promoting the digestion and absorption of host nutrients, maintaining normal physiological functions of the intestine, regulating body immunity, and antagonizing the colonization of pathogenic microbes [17]. Changes in the intestinal environment disrupt the homeostasis of the intestinal flora, causing autoimmune diseases, such as inflammatory bowel diseases, type I diabetes, cardiovascular diseases, central nervous system diseases, allergic diseases, rheumatoid arthritis, and systemic lupus erythematosus [18].

In addition, the oral microbial flora comprises one of the most complex microbial communities known. Microbes in the mouth and intestine interact with each other to some extent. Bacteria in the mouth enter the stomach by food ingestion, eventually reaching the intestine. At least 700 microbe species have been identified to occur in the oral bacterial community or biofilms [16,17]. Similar to intestinal flora, a disruption in oral flora is closely associated with the occurrence of many malignant tumors and autoimmune diseases.

It is now well established that the interaction between IgA and the microbiota promotes homeostasis with the host to prevent disease [19]. Moreover, intestinal microbiota and its metabolites play a key role in IgA-immune responses [20]. The SIgA produced independently of T-cell selection are considered natural or innate antibodies that act through an endogenous mechanism driving homeostatic production of polyreactive IgA with innate specificity to microbiota. So-called innate SIgA, thus, coats a subset of commensals [20]. In contrast, following stimulation by pathogens or vaccines adaptive-IgA is selected in T cell-dependent germinal centers generating advanced mutualism with the microbiota by selecting and diversifying beneficial microbial communities [21]. Both types of SIgA are effective at excluding microorganisms from the gut invasion.

## 2. Pathophysiology of IgA Nephropathy

IgA nephropathy (IgAN) is defined as a primary chronic mesangial proliferative nephritis with predominant deposition of IgA [22]. Diagnosis of IgAN requires proof of glomerular IgA deposition in a kidney biopsy. It can be associated with IgG and C3 deposits. However, IgA deposition in the mesangial area is observed in 5–15% of autopsied cases without any renal disease and about 20% of donated kidneys without abnormal urinary findings [23]. Thus, the presence of glomerular IgA deposition is not always pathological. Several studies since the 80′s revealed that IgAN is associated with the presence of immune complexes in the circulation [23]. Experimental models were crucial to demonstrate the role of IgA-IC in IgAN. Rifai et al. [24] described the first animal model of IgAN in 1979. Using murine anti-dinitrophenol (DNP) and DNP-conjugated bovine serum albumin (DNP-BSA), they generated circulating IgA-IC and demonstrated that these complexes were prone to mesangial deposition. They found that for mesangial deposition to occur IgA-IC needed to be either administered repeatedly or to be present persistently in the circulation. The same group also demonstrated the importance of IgA-IC size in mesangial deposition by studying different pIgA-antigen complexes [25]. In the early 1980s, Isaacs et al. confirmed the importance of circulating IgA-IC in mesangial IgA deposition and initiation of glomerular injury. In these studies, animals were immunized with a bacterial-derived polysaccharide or chemically-modified dextran [26]. These studies emphasized both the importance of continual IgA-IC formation as a driver for mesangial IgA deposition and progression of IgAN, and the critical role played by polymeric IgA in the formation of circulating nephritogenic IgA-IC.

To assess characteristics of nephritogenic IgA and IgA-IC which have high affinity to glomerular mesangium and in an attempt to discriminate why not all IgA-IC (for example IgA rheumatoid factors) can induce IgAN, two groups analyzed the physicochemical properties of mesangial IgA deposits from patient biopsies [27,28]. Mesangial IgA deposits were found to be composed of mainly dimeric IgA with a negative charge [28]. Later on, with the discovery of aberrant hinge glycosylation of circulating IgA1 in IgAN patients, notably with a galactose deficiency (Gd-IgA1) [29], it was shown that mesangial deposits composed of Gd-IgA1 which may explain their negative charge [30]. Gd-IgA1 was shown to be immunogenic generating autoantibodies to free N-acetylgalactosamine (GalNac) residues, which were mainly of IgG class [31]. Recently, it has been shown that IgG anti-Gd-IgA1 can be found in the mesangium of some patients with IgAN [32]. These data suggest that Gd-IgA1-IC is nephrotoxic.

The origin of anti-Gd-IgA1 autoantibodies remains unknown in IgAN. It is likely that the lack of galactose residues unmask immunogenic epitopes on the IgA1 hinge region generating autoimmunity. A recent GWAS study, however, identified galactose transferase candidate genes as novel insights into the genetic regulation of O-glycosylation in IgAN [33,34]. Polymorphisms of C1GALT1 enzymes have been described that may affect O-glycosylation of IgA1 in IgA-secreting plasma cells of patients with IgAN [35]. Whether altered galactosylation processes result from immunometabolic signals emanating from gut microbiota remains also unknown. One important point is that in any case altered glycosylation in the hinge region of IgA1 favors interaction with a mesangial IgA1 receptor, CD71 [36].

Another major component of nephrotoxic IgA-IC is the soluble form of the IgA Fc receptor, soluble CD89 (sCD89). CD89 has been found to be activated by IgA-IC playing a role in the pathogenesis of IgAN [37,38]. Importantly, CD89 is shed from the cell surface of blood monocyte from patients with IgAN generating soluble form of CD89 (sCD89) which has not been found in healthy individuals [39,40]. Unexpectedly, the sCD89 appears in the circulation as part of the IgA-IC [40], whereas it was lost in patients with progressive IgAN indicating a “trapping” within the mesangium [41]. Studies with animals expressing human IgA1 with or without CD89 demonstrated experimentally that sCD89 is required for mesangial IgA1 deposit formation inducing renal dysfunction [42]. Later, studies in patients revealed the presence of the mesangial sCD89 deposits in recurrent IgAN after transplantation [43] and in children with IgAN [44]. In the latter study, sCD89 was found in the mesangium of patients both free and colocalizing with IgA [44]. Interestingly, sCD89 was found to be responsible for mesangial cell proliferation in a receptor-mediated manner involving the transferrin receptor mTOR pathway [44]. CD89, thus seems to play a major role in the pathogenesis and progression of IgAN.

## 3. Evidence for the Role of a Gut-Kidney Inflammatory Axis in IgAN

The role of mucosal immunity in the development of IgAN has been explored since the 80′s [45]. Genome-wide association studies (GWAS) have recently revealed candidate regions (loci) in the genome that were associated with the risk of developing the disease [46]. From this large study, a genetic risk for IgAN was identified from a relationship between the genetic probability of developing IgAN and local pathogen diversity including viruses, bacteria, protozoa, and helminths. IgAN susceptibility loci were related to the predisposition to inflammatory bowel diseases (IBD), with the genes implicated in the protection of the gut integrity and mucosal immune response towards the gut environment. The conclusion was that the striking association between genetics and environmental elements could induce functional changes in the gut mucosal immune system favoring the onset of the disease.

Gut abnormalities have been described in IgAN patients, such as altered gut permeability [47,48,49]. More importantly, recently it has been reported in a large study of around 4000 Swedish patients with IgAN display an increased risk of inflammatory bowel disease (IBD) both before and after their disease diagnosis. Moreover, IBD elevates the risk of IgAN progression to end-stage renal disease (ESRD) [50]. These elements suggested the possible benefits of the glucocorticosteroid budesonide, a drug used mainly in IBD that targets the intestinal immunity and local inflammation in the gut mucosa and Peyer patches. A phase 2b trial was conducted using a double-blind, randomized, ileum targeting release formulation of the budesonide (TRF budesonide; Nefecon™). This formulation allowed delivery of the drug to the upper ileum (where PPs are mostly represented) and acts on the local immune hyperresponsiveness, reducing adverse systemic effects (only 10% reaches systemic circulation) [51]. TRF-budesonide (16 mg/day) reduced proteinuria and stabilized renal function in patients with IgAN justifying a phase 3 trial. These results indicate that gut mucosal inflammation is involved in the IgAN pathology.

## 4. Role of Food Antigens and Infections in IgAN

Mucosal antigenic exposure is able to form circulating IgA-IC and the development of IgA-IC mediated glomerulonephritis in mice [52]. As high serum levels of IgA anti-gliadin have been reported in patients with IgAN [53], a study addressing gluten sensitivity in IgAN patients (without signs of coeliac disease) identified a rectal mucosal sensitivity to gliadin, a lectin present in gluten, suggesting sub-clinical inflammation to gluten might be involved in the pathogenesis of IgAN [54]. The introduction of gluten was able to induce an IgAN phenotype in a gluten-free α1KICD89Tg mouse model of IgAN obtained after three generations under a gluten-free diet [55]. Gliadin was able to interact not only with IgA antibodies but also with sCD89 and the CD71 IgA1 mesangial receptor [55]. Gluten depletion from the diet was able to prevent the disease in mice [55] providing an explanation for decreased proteinuria observed in patients during treatment with a gluten-free diet [56]. Interestingly, half of Swedish IgAN patients have also a rectal mucosal sensitivity to soy or cow’s milk suggesting that immune reactivity towards a broad pattern of food antigens may also play a role in the pathogenesis of IgAN [57]. In addition to intrinsic food antigens, food-borne-microbial contaminants may also provide an antigenic stimulus in IgAN [58].

Mucosal infections, such as upper respiratory tract infections, are associated with macroscopic hematuria in IgAN which is highly suggestive of a role of the mucosal immune system in this disease [59]. Some authors reported that glomerulonephritis is associated with *Staphylococcus aureus* infection in IgAN patients [60,61]. The latter studies were able to induce IgAN in mice immunized subcutaneously with *Staphylococcus aureus* antigens [62]. Other groups have also demonstrated the development of experimental IgAN following oral immunization with *Haemophilus parainfluenzae* antigens and associated with glomerular deposition of outer membranes of these antigens [63,64]. In addition to the bacterial infection, viral infection also links to IgAN-like nephropathies, such as Aleutian disease by parvovirus infection or mucosal viral infection with Sendai virus [65,66]. The mechanisms by which such microorganisms are able to induce the disease were shown to be mediated by Toll-like receptors (TLR) notably TLR9 which was correlated with disease severity [67]. These studies suggest that exogenous microorganism antigens could play a key role in the development and progression of IgAN.

## 5. Which Role for Gut Microbial Dysbiosis in IgAN?

Although altered permeability of the gut epithelium has been described in IgAN patients since the 80′s [47,48,49], the first evidence for altered microbiota came recently from an experimental model, following overexpression of B-cell activating factor (BAFF) [68]. These mice develop B cell hyperplasia which was associated with high production of polymeric hypoglycosylated mouse IgA and IgA deposits in the glomeruli associated with IgM. Interestingly, IgAN-like development of the disease was dependent on the microbiota suggesting that overexpression of this cytokine could promote dysbiosis. However, how dysbiosis induces an IgAN phenotype remains unclear. These results led Gesualdo et colleagues to conduct microbiota analysis in an Italian cohort of IgAN patients as compared to healthy individuals [69]. Interestingly, patients with the progressive disease showed the lowest microbial diversity compared to non-progressors and healthy individuals. This study revealed an increase in Firmicutes in the stool samples of both types of IgAN patients due to a high percentage of Ruminococcaceae, Lachnospiraceae, Eubacteriaceae, and Streptococcaeae as genera/species. This was contrasting with *Clostridium*, *Enterococcus* and *Lactobacillus* genera observed in stools from healthy individuals. Although the latter study lacks a control group with different types of chronic kidney diseases (CKD), which is known to alter the microbiota due to uremia [70], recently another group have also found microbiota dysbiosis in IgAN patients when compared with membranous nephropathy [71]. The abundance of Escherichia-Shigella, which was consistent with the previous study, and Defluviitaleaceae incertae sedis were higher in IgAN than those in healthy individuals. However, lower levels were found for *Roseburia*, *Lachnospiraceae*, *Clostridium sensu stricto*, *Haemophilus*, and *Fusobacterium*. Interestingly, the level of *Megasphaera* and *Bilophila* was higher in IgAN patients, whereas that of *Megamonas*, *Veillonella*, *Klebsiella* and *Streptococcus* was lower compared with those with membranous nephropathy. In this study, *Prevotella* levels were positively correlated with the level of serum albumin, while *Klebsiella*, *Citrobacter* and *Fusobacterium* were negatively linked. Interesting, a positive correlation was shown between *Bilophila* and the presence of glomerular crescents based on the Oxford classification of IgAN.

Altered mucosal immunity may also have a role in the development of the disease [72]. In this study, the authors showed that IgAN patients displayed circulating gut-homing (CCR9+ β7 integrin+) regulatory B cells and IgA+ memory B cells that may favor the hyper synthesis of IgA in mucosal sites [72]. Moreover, increased serum levels of BAFF were positively correlated with amounts of five specific microbiota metabolites (4-1,1,3,3-tetramethylbutyl phenol, p-tert-butyl-phenol, methyl neopentyl phthalic acid, hexadecyl ester benzoic acid, and furanone A). Phenol has a toxic effect on the gut by reducing barrier function, increasing gut permeability, and inducing mucosal hyperresponsivity [73]. Increased permeability found in IgAN patients [47,48,49] could favor microbiome products, such as lipopolysaccharide (LPS) and lipoteichoic acid which can activate the GALT through TLR pathways. Increased TLR4 and TLR9 expression by blood leukocytes have been indeed observed in IgAN patients [67,74]. Moreover, a polymorphism at the -159 site (T to C) of CD14, a member of TLR4 implicated in LPS binding, was found to be correlated with progressive IgAN [75]. Moreover, recently it has been shown that oropharyngeal pathobionts, such as *Neisseria* are found in IgAN patients associated with elevated Neisseria-targeted serum IgA antibodies [76]. These authors translated their findings in an experimental IgAN model driven by BAFF overexpression in BAFF-transgenic mice, rendered susceptible to *Neisseria* infection by the introduction of a humanized CEACAM-1 transgene. Colonization with *Neisseria* augmented levels of systemic *Neisseria*-specific IgA that were also found in the kidneys.

Additional evidence for the role of microbiota in IgAN comes from a humanized mouse model of IgAN, the α1KICD89Tg mice. These mice express the human heavy chain of IgA1 and the human CD89 IgA Fc receptor. These animals spontaneously develop IgAN characterized by human IgA1 and mouse C3 mesangial deposits associated with hematuria and proteinuria [42,77]. As germ-free housing impairs IgA1 production in α1KI animals [78], the strategy used was an intervention targeting the gut microbiota by broad antibiotics in 8 or 12 week-old animals, the age when IgA1 reached expected serum levels and IgA1 deposits were clearly detected followed by proteinuria [42]. Commensal flora depletion by broad-spectrum antibiotics (vancomycin/amoxicillin/neomycin/metronidazole) was established after 8 weeks [79]. This treatment prevented the development of mesangial IgA1 deposits and proteinuria. Importantly, similar treatment with broad-spectrum antibiotics reverted established disease (12 to 16 week-old animals) by completely abolishing IgA1 mesangial deposits and reverting proteinuria. This was associated with decreased concentration of human (h) IgA1-mouse (m) IgG immunocomplexes in the circulation. However, surprisingly, such treatment did not affect total IgA1 levels in the circulation nor levels of IgA+ B cells in the gut submucosa strongly suggesting that nephrotoxic IgA1 originated from the gut. One could thus postulate that qualitative differences, such as altered glycosylation may take place in the gut inducing the detected hIgA1-mIgG immune complexes. Together, these data provide further evidence that a gut-kidney axis may play a crucial role in IgAN.

## 6. Therapeutic Approaches Targeting the Microbiota

As microbiota dysbiosis seems to be associated with the disease progression [69], one could propose that antibiotics could be used to treat or prevent IgAN development or progression. However, given the risk of collateral effects of broad antibiotic therapy, such as *Clostridium difficile* infections and antibiotic resistance, it seems reasonable to search specific commensals or pathobionts to be targeted by single drugs without modifying the whole microbiome. In this line, a study with rifaximin, a non-absorbable oral antibiotic, was able to induce a modulation of the gut microbiota, favoring the growth of bacterial species beneficial to the host [80]. Rifaximin treatment in the α1KICD89Tg mice decreased the urinary protein-to-creatinine ratio, serum levels of IgA1-sCD89 and mIgG-hIgA1 complexes, IgA1 mesangial deposition, and inflammatory cells (CD11b+) in the kidney. Moreover, rifaximin treatment significantly decreased BAFF, pIgR and TNF-α mRNA expression. This pre-clinical data reveals a possible place for rifaximin in future clinical trials for this disease.

As demonstrated in our pre-clinical studies, dietary interventions, such as a gluten-free diet seem promising in IgAN and should be considered despite the absence of controlled clinical trials. Notably, patients with having anti-gliadin serum antibodies may benefit from this approach.

The use of prebiotics, probiotics, or symbiotics displays anti-inflammatory, anti-oxidative, and other favorable gut-modulating properties [81]. Species from the Lactobacilli and Bifidobacteria genera were shown to improve humoral immune responses against toxins and antigens. Probiotics or symbiotics have been used in patients with CKD with beneficial effects on the uremic toxins, oxidative stress, and inflammation [82]. Interestingly, it has been shown at the pre-clinical level that the administration of *Saccharomyces boulardii* is able to decrease intestinal inflammation, reduced systemic IgA response and prevented oral-poliovirus vaccine-induced IgAN in mice [83].

Gut microbiota manipulation appears to be a new option for therapy of many diseases including dietary interventions, prebiotics, and probiotics, or through fecal microbiota transplantation (FMT). The latter approach has been used successfully for the treatment of patients with *Clostridium difficile* infection [84]. We have recently performed a collaborative study with the Gesualdo group examining the FMT approach in antibiotic-treated α1KICD89Tg animals using stools from healthy controls, non-progressor, and progressor IgAN patients by oral gavage [85]. Patient-derived FMT was clearly able to modulate renal phenotype and inflammation in mice. Microbiota from patients with the progressive disease was able to induce an increase of serum BAFF and Gd-IgA1 levels which were associated with soluble CD89 and IgA1 mesangial deposits. On the other hand, the microbiota from healthy individuals was able to induce a reduction of albuminuria immediately after gavage, restoring CD89 full expression on blood cells and decreasing inflammation in the kidney. The main bacterial phyla composition and volatile organic compounds profile significantly differed in mouse gut microbiota between FMT groups. This pre-clinical study suggests that microbiota modulation by FMT from healthy individuals downregulates IgAN phenotype and opens new strategies for therapeutic approaches in IgAN. This is in agreement with data from others showing that in patients suffering from CKD, FMT has a positive action on intestinal microbiota diversity limiting the increase of uremic toxins released from the gut cresol pathway [86]. This pathway is characterized by the production of an aromatic compound, the 4-cresol (i.e., p-cresol or p-methylphenol), a synthetic precursor for manufacturing a great variety of chemical products including synthetic resins, disinfectants, antioxidants, preservatives, fumigants, explosives, and others. In nature, 4-cresol is generated by anaerobic bacteria as a byproduct during the metabolism of phenylalanine and tyrosine [87]. Although there are no data so far for FMT-based treatment in IgAN, an interventional study (clinical trial NCT03633864) is currently being conducted in China and intends to define the safety and efficiency of FMT in IgAN subjects resistant to the standard therapy.

## 7. Conclusions and Hypotheses

Taken together, it appears clear that a pathogenic gut-kidney axis is operating in IgAN. We formulate two alternative hypotheses (Figure 1): 1/Genetically susceptible individuals for gut inflammation as shown on risk loci in GWAS studies would generate microbial dysbiosis upon exposure to certain environmental factors or infectious agents. Both triggers could induce a mis-galactosylation of IgA1 antibodies. This hypothesis is supported by the BAFF-transgenic model which develops altered IgA glycosylation and mesangial IgA deposits in mice [68]. In patients, elevated levels of BAFF cytokine in serum were found to be linked to specific microbiota metabolites [72]. Gut dysbiosis could drive aberrant IgA glycosylation through inhibition of COSMC expression, a molecular chaperone required for expression of active T-synthase, the only enzyme that galactosylates the Tn antigen (GalNAcα1-Ser/Thr-R) to form core 1 Galβ1–3GalNAcα1-Ser/Thr (T antigen) during mucin type O-glycan biosynthesis [88] or by pro-inflammatory cytokines altering C1GalT1 and ST6GalNAc-II enzymes [89]. 2/Microbial dysbiosis due to environmental factors or infectious agents could induce gut inflammation and lead to altered gut permeability as shown in patients [47,48,49] resulting in the formation of pathogenic IgA1 immune complexes. This is supported by the disappearance of mesangial IgA1 deposits in α1KICD89Tg mice after giving a gluten-free diet for three generations [55] or by treating these mice with antibiotics [79]. There is thus a possibility that dysregulation of the microbiota caused by a change in diet or infections leads to gut inflammation and then the production of mis-galactosylated IgA1. This highlights a possible role for the dysregulation of the interplay between the intestinal immunity, diet and microbiota in the onset of the disease. However, further work is needed to delineate the mechanisms by which nephrotoxic Gd-IgA1 immune complexes are generated following interactions between the microbiota and mucosal IgA system in the pathogenesis of IgAN.

## Figures and Tables

**Figure 1 microorganisms-10-00683-f001:**
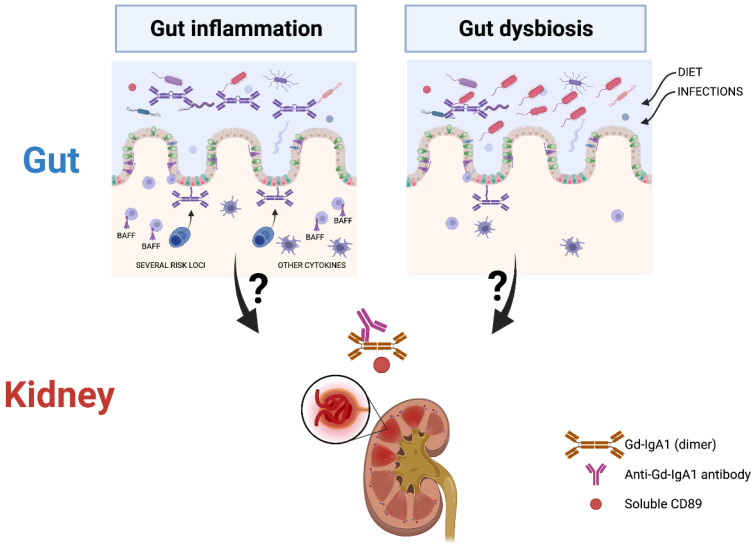
Putative gut origin of nephrotoxic Gd-IgA1 in IgA nephropathy.The left side shows gut inflammation based on genome-wide association studies (GWAS) studies indicating candidate genes involved in mucosal immunity together with the role of B-cell activating factor (BAFF) in IgAN. The right side illustrates dysbiosis as a primary event in: Immunoglobulin A nephropathy (IgAN). Nephrotoxic Gd-IgA1-immune complexes are indicated (GdI-gA1, Anti-Gd-IgA1 antibody and soluble CD89). The figure was prepared using a Biorender software license.

## Data Availability

Not applicable.

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
