# Peer review of "Is There a Role for Gut Microbiome Dysbiosis in IgA Nephropathy?"

_microorganisms, 2022, doi:10.3390/microorganisms10040683_

Round 1

Reviewer 1 Report

This is an interesting, generally well-written, novel, comprehensive and timely review that discusses IgA nephropathy (IgAN), including its possible microbiota-driven pathogenesis. The major areas of interest pertaining to the pathobiology of IgAN are properly covered. Perhaps, authors could elaborate a bit more as to why and how IgAN patients develop autoimmune IgG responses to hypo-galactosylated IgA1. Does the lack of galactosamine residues unmask immunogenic epitopes on IgA1? Do these anti-IgA1 responses always target the hinge region of IgA1? I also wonder whether galactose depletion involves O-linked glycans on the hinge region of IgA1. Do these galactose-containing glycans serve any specific function within the long hinge region of IgA1? Are they needed for IgA1 binding to CD71? Finally, do IgA-secreting plasma cells specifically express galactose transferases or this expression already occurs in IgA1 class-switched B cells? Is this expression regulated by immunometabolic signals emanating from the gut microbiota? The following are additional (minor) comments to enhance the manuscript.

Line 18: “FMT” should be spelled out.

Line 36: please, add the “pIgR” acronym after mentioning the polymeric Ig receptor. This acronym appears later in the text without being properly spelled out (e.g., line 298).

Lines 52-54: this sentence needs some editing.

Line 62: please, replace “neutrophiles” with neutrophils.

Line 88: a more recent reference may be needed (e.g., Chen et al. Nat. Rev. Immunol. 2020). The involvement of BAFF/APRIL in intestinal T cell-independent IgA production has been formally demonstrated in vivo by a recently published study (Grasset EK et al. Sci. Immunol. 2020).

Line 151: “GalNa”  is not spelled out.

Line 209: please, revisit the following sentence “… suggesting immune reactivity a broad pattern of food antigens …”.

Line 225: “Taking together …” should be corrected.

Line 235: “… the mechanism by how …” should be corrected.

Lines 243 and 317: “CKD” is not spelled out.

Line 256: the sentence “An additional study from has shown that …” needs to be amended.

Line 260: “BAFF” should be spelled out. Its function should be briefly described.

Lines 264-266: this sentence is missing and needs to be thoroughly edited.

Line 266: : “TLR” should be spelled out. The function of these innate receptors should be briefly described to the benefit of a broad audience.

Line 272: please, spell out and explain “1KICD89Tg mice”, to the benefit of a broad audience.

Line 298: “pIgR” should be spelled out. Its function should be briefly described.

Lines 303 and 305: FMT is spelled out twice.

Line 319: what is the “gut cresol pathway”? Please, explain.

Lines 332-334: “COSMC”, “C1GalT1” and “ST6GalNAc-II” should be spelled out and defined.

Author Response

We thank the reviewer for her/his comments. All comments have been addressed in the revised version.

Reviewer 2 Report

The paper presents an extensive review on gut microbiome dysbiosis in IgAN, it’s a topic of interest to the researchers in the related areas, and the detailed comments are as follows:

  • It is noted that your manuscript needs more subtitles to identify the issues you talk about. Especially in the fifth part, you can list the treatment-related content separately.   
  • As for the new therapeutic strategies you mentioned in your manuscript, the dietary interventions, prebiotics, probiotics need further description. Whether there are clinical trials and the major conclusions of these trials should be in the review.
  • In section 5, the first sentence of the second paragraph may lack some elements.

Author Response

(The authors gave the same response as above.)
